# The Transcription Factors *TaTDRL* and *TaMYB103* Synergistically Activate the Expression of *TAA1a* in Wheat, Which Positively Regulates the Development of Microspore in *Arabidopsis*

**DOI:** 10.3390/ijms23147996

**Published:** 2022-07-20

**Authors:** Baolin Wu, Yu Xia, Gaisheng Zhang, Junwei Wang, Shoucai Ma, Yulong Song, Zhiquan Yang, Elizabeth S. Dennis, Na Niu

**Affiliations:** 1Key Laboratory of Crop Heterosis of Shaanxi Province, Wheat Breeding Engineering Research Center, Ministry of Education, College of Agronomy, Northwest A&F University, Yangling 712100, China; wbl97@nwafu.edu.cn (B.W.); xiayu0325@sjtu.edu.cn (Y.X.); zgs02196@163.com (G.Z.); wjw@nwsuaf.edu.cn (J.W.); mashoucai@sohu.com (S.M.); sylbl1986@nwsuaf.edu.cn (Y.S.); zhqyang6688@nwsuaf.edu.cn (Z.Y.); 2School of Agriculture and Biology, Shanghai Jiao Tong University, Shanghai 200240, China; 3Agriculture and Food, Commonwealth Scientifc Industrial Research Organisation, Canberra, ACT 2601, Australia

**Keywords:** wheat, male sterility, pollen, *TAA1a* promoter, transcriptional regulation

## Abstract

Pollen fertility plays an important role in the application of heterosis in wheat (*Triticum aestivum* L.). However, the key genes and mechanisms underlying pollen abortion in K-type male sterility remain unclear. *TAA1a* is an essential gene for pollen development in wheat. Here, we explored the mechanism involved in its transcriptional regulation during pollen development, focusing on a 1315-bp promoter region. Several *cis*-acting elements were identified in the *TAA1a* promoter, including binding motifs for *Arabidopsis thaliana* AtAMS and AtMYB103 (CANNTG and CCAACC, respectively). Evolutionary analysis indicated that *TaTDRL* and *TaMYB103* were the *T. aestivum* homologs of *AtAMS* and *AtMYB103*, respectively, and encoded nucleus-localized transcription factors containing 557 and 352 amino acids, respectively. *TaTDRL* and *TaMYB103* were specifically expressed in wheat anthers, and their expression levels were highest in the early uninucleate stage; this expression pattern was consistent with that of *TAA1a*. Meanwhile, we found that TaTDRL and TaMYB03 directly interacted, as evidenced by yeast two-hybrid and bimolecular fluorescence complementation assays, while yeast one-hybrid and dual-luciferase assays revealed that both TaTDRL and TaMYB103 could bind the *TAA1a* promoter and synergistically increase its transcriptional activity. Furthermore, TaTDRL-EAR and TaMYB103-EAR transgenic *Arabidopsis* plants displayed abnormal microspore morphology, reduced pollen viability, and lowered seed setting rates. Additionally, the expression of *AtMS2*, a *TAA1a* homolog, was significantly lower in the two repressor lines than in the corresponding overexpression lines or WT plants. In summary, we identified a potential transcriptional regulatory mechanism associated with wheat pollen development.

## 1. Introduction

Wheat (*Triticum aestivum* L.) is an important cereal crop worldwide [1]. This crop feeds nearly 40% of the world’s population, contributing to approximately 20% of the global total caloric intake [2,3,4]. Demand for wheat is expected to grow as the world’s population increases; however, the amount of arable land available for growing food crops is limited [5,6], and increasing the yield per unit area of wheat remains a fundamental goal in wheat breeding. Numerous studies have confirmed that better breeding methods can effectively improve grain quality, increase crop yield, and allow for better adaption to adverse environments [7,8].

Heterosis refers to a phenomenon in which the F1 generation of genetically diverse individuals of the same species is superior to the parents in aspects such as yield, stress resistance, and adaptability [9,10]. Accordingly, the utilization of heterosis in wheat breeding represents one of the most effective strategies for improving the yield of this crop [11,12].

Plant male sterility (MS) represents a valuable and widely used trait for the investigation and utilization of crop heterosis. Male sterility in plants is often associated with abnormal stamen meiosis, callose metabolism, tapetal development, pollen wall development, and cracking [13]. For example, in the rice (*Oryza sativa*) PTGMS (Photo-thermo-sensitive genic male sterility) line 95850, sporopollen cannot properly accumulate, resulting in pollen wall thinning and pollen rupture [14]. The pollen of the 1740CA sterile line of *Brassica napus* has a very thin outermost pollen wall layer, resulting in cell deformation and pollen abortion [15]. The male-sterile line 64A of radish (*Raphanus satious* L.) aborts in autumn and winter when vacuoles appear in its mononuclear microspores, and the pollen exine is thin [16]. The sporopollenin of the pollen wall of the RC7 sterile line of Chinese cabbage is not continuously deposited with low content, and the pollen exine does not form normally, resulting in pollen abortion [17]. Many vesicles appear in the pollen wall of the K-77(2) sterility line of wheat [18]. Isonuclear alloplasmic CMS (Cytoplasmic male sterility) lines produced by different cytoplasmic types of *Triticum* and *Aegilops* have significantly thinner pollen walls than normal, the pollen shrinks significantly, and this eventually leads to male sterility [19]. Therefore, in both monocotyledons and dicotyledons, abnormal pollen wall development and abnormal microspore development are important factors leading to male sterility.

The pollen exine is mainly composed of sporopollenin, and the synthesis of sporopollenin is the result of the joint participation of tapetal cells and microspores. Sporopollenin is difficult to decompose, and is characterized by resistance to oxidation and strong acids and bases and insolubility in organic and inorganic acids, inorganic salts, and lipid solvents [20]. Given these properties, sporopollenin is effective at protecting pollen from the external environment and ensuring normal activity [21,22,23]. Sporopollenin is mainly composed of long-chain fatty acids and their derivatives [24], and its biosynthesis is closely related to fatty acid metabolism [25]. In *Arabidopsis thaliana*, Acyl-COA synthetase 5 (ACOS5) is specifically expressed in the anther tapetal tissue, and has strong specificity for oleic acid (C18:1). This suggests that *ACOS5* is involved in the metabolism of medium chain fatty acids (C8 to C16) related to sporopollenin synthesis during anther development; mutation of this gene leads to the inhibition of the sporopollenin synthesis pathway, and the loss of the microspore outer wall leads to pollen abortion [26,27,28]. *CYP703A2* encodes a cytochrome P450 hydroxylase that catalyzes the formation of long-chain fatty acids into monohydroxy fatty acids, and the knockout of this gene leads to the loss of pollen exine and partial male sterility [23]. In rice, *CYP704B2* encodes a cytochrome P450 that was shown to catalyze the hydroxylation of C16 and C18 fatty acids, thereby regulating the synthesis of pre-sporopollenin substances, and ultimately affects the formation of pollen exines [29]. The *A*. *thaliana MALE STERILITY 2* (*MS2*) gene encodes a lipoyl reductase that is specifically expressed in the tapetum during anaphase of meiosis. The pollen wall of the *ms2* mutant is very thin and the *ms2* mutant is male-sterile [30,31,32]. The *OsMS2* gene of rice is homologous to the *MS2* gene of *A*. *thaliana*. *OsMS2* also encodes a fatty acyl reductase that is involved in the synthesis of fatty alcohols required for pollen exine development. Mutations in the *OsMS2* gene result in microspores that do not form a complete pollen wall structure, ultimately leading to microspore abortion [33]. The wheat *TAA1* gene (*T. aestivum anther 1*) is highly homologous and functionally similar to *MS2*, and its protein product can reduce C14:0, C16:0, and C18:1 fatty acyl-COAs to fatty alcohols [30,34]. These results suggest that fatty alcohols are important for pollen wall development in both monocotyledons and dicotyledons, and that the underlying molecular mechanism may be conserved in plants.

Abundant evidence supports that abnormal tapetal development has an important and negative influence on the development of pollen exine, and can result in sterile pollen [35,36,37]. Tapetal cells in wheat and rice contain a structure called the Ubisch body that secretes pre-sporopollenin [38]. Mutations in the transcription factors *ABORTED MICROSPORES* (*AtAMS*) and *AtMYB103* of *A. thaliana* cause the abnormal degradation of the tapetum [35]. Related studies have shown that *AtMYB103* and *AtAMS* regulate sporopollenin biosynthesis in the tapetum by synergically activating *CYP703A2* [39]. In the tapetum, *AtAMS* directly interacts with and regulates *AtMYB103*, forming a feed-forward loop between the two, thereby promoting rapid sporopollenin synthesis [40]. The rice homolog of *AtAMS*, *TAPETUM DEGENERATION RETARDATION* (*OsTDR*), regulates pollen exine development by directly controlling *OsC6* expression [41].

Wheat is known to contain three apparently homologous *TAA1* genes (*TAA1a*, *TAA1b*, and *TAA1c*). The TAA1a protein has fatty alcohol-producing, fatty acyl-CoA reductase activity and can generate long- and very-long-chain fatty alcohols. TAA1a plays a role in tapetal cells and, consequently, is important for pollen fertility. The abnormal expression of the *TAA1a* gene leads to impaired formation of the pollen wall, eventually resulting in microspore rupture and degeneration and, consequently, male sterility [34]. However, the transcriptional regulatory mechanisms and associated promoter region underlying the role of *TAA1a* in the regulation of pollen fertility remain unclear. We have previously used the MS(KOTS)-90-110 male-sterile line (In subsequent text, it is represented by MS) as the female parent in crossings with other fine wheat varieties. This line has the advantages of stable sterility, wide-ranging recovery sources, and a small phenotypic coefficient of variation in the F1 generation [42], resulting in strong and dominant hybrid wheat combinations. In the previous research work of our laboratory, the K-type cytoplasmic male sterile line (MS) was crossed with the restorer lines with high restoring degree to obtain the F1 generation, and the sterile line was selected as the recurrent parent, and the fertile plants in the F1 generation and backcross progeny were selected for successive generations of backcross to produce a fertile near-isogenic line. Represented by MR(KOTS-90-110) (In the following text, represented by MF). Its cytoplasmic background is the cytoplasm of *Aegilops Kotschyi*. The comprehensive investigation of cytology and phenotype shows that they have no difference in other genetic backgrounds and agronomic traits except fertility and sterility. This is excellent material for studying the sterility mechanism of MS (KOTs)-90-110 (MS) [43,44]. Consequently, in this study, we employed the MS(KOTS-90-110) line and MR(KOTS-90-110) to explore the mechanism involved in the transcriptional regulation of the *TAA1a* gene during pollen development using microspore abortion as a readout.

Here, a 1315-bp *TAA1a* promoter sequence (*proTAA1a*) was cloned from the fertile near-isogenic line MR(KOTS-90-110). By analyzing the cis-acting elements of *proTAA1a*, we identified *TaTDRL* and *TaMYB103* as candidate transcription factors regulating *TAA1a*. Furthermore, yeast one-hybrid and dual-luciferase assays demonstrated that TaTDRL and TaMYB103 cooperated to synergistically activate the *TAA1a* promoter. Further investigation of the function of TaTDRL and TaMYB103 using TaTDRL-EAR and TaMYB103-EAR transgenic *Arabidopsis* plants generated through chimeric repressor silencing technology (CRES-T) [45] indicated that both TaTDRL and TaMYB103 play an active regulatory role in pollen development in transgenic *Arabidopsis*. Combined, our findings indicated that TaTDRL may interact with TaMYB103 at the *TAA1a* promoter, thereby exerting a regulatory effect on *TAA1a* expression, and consequently, pollen exine and microspore development in wheat.

## 2. Results

### 2.1. Characterization of Abnormal Anther and Microspore Development in the CMS Line

To further clarify the characteristics of CMS-related anther abortion, we assessed microspore morphology in different stages of anther development. Carmine acetate dyeing results indicated that the microspores of the MF line in the tetrad, early uninucleate, late uninucleate, binucleate, and trinucleate stages were of normal morphology. In each stage, the nucleus of the microspore was clearly visible, and the microspore was regular, round, filled with cytoplasmic contents, and thick (Figure 1A–E). Microspore development in the MS(KOTs)-90-110 line did not differ from that of its isogenic line in the tetrad and early-uninucleate stages (Figure 1I,J). However, in the late uninucleate stage, the microspores of the CMS line displayed plasmolysis and a shriveled morphology (Figure 1K). In the binucleate stage, meanwhile, the microspores had diffuse nuclei and a lack of intracellular contents (Figure 1L). The microspores of the CMS line failed to develop to the trinucleate stage; at this time, the cells were thin and of irregular shape, the nuclei were heavily dispersed, and the cells were withered and almost devoid of content (Figure 1M). Additionally, unlike that seen with the MF line, the anthers of the CMS line could not dehisce (Figure 1F,N). Iodine–potassium iodide (I_2_–KI) staining showed that the mature pollen grains of the MF line were full, regular, and dyed black (Figure 1G); however, the pollen cells of the CMS line were wrinkled, irregular, and the staining was a brown color and uneven (Figure 1O), indicating an almost complete lack of starch accumulation in the microspores. Scanning electron microscopy (SEM) further showed that the pollen grains of the CMS line had an abnormal, depressed morphology at the trinucleate stage (Figure 1H,P).

### 2.2. Isolation and Sequence Analysis of the TAA1a Promoter

The 1315-bp promoter sequence of *TAA1a* (*proTAA1a*) was isolated by PCR from the genomic DNA of the fertile near-isogenic line MR(KOTS)-90-110 (MF). *Cis*-acting elements identified in *proTAA1a*, in addition to core elements such as the TATA-box and CAAT-box, included 10 light-responsive elements, five plant hormone response elements, three pollen-specific elements, eight stress response elements, seven E-boxes (CANNTG; N = A/T/C/G), and one MYB103-specific recognition motif (CCAACC) (Appendix A).

### 2.3. Expression Patterns of the Transcription Factors TaTDRL, TaMYB03, and TAA1a in Wheat Anthers

In wheat, the *TAA1a* gene is expressed in an anther-specific manner [34]. Here, we assessed the relative expression levels of *TAA1a* (*Traes*CS4B02G363100.1) in various tissues and stages of anther development in the MF and MS lines using qRT-PCR. As shown in Figure 2A, the highest *TAA1a* expression levels were detected in anthers (Figure 2A), which was consistent with that reported by Wang et al. The expression level of the *TAA1a* gene in the MF line was highest in the late mononuclear stage. Additionally, the expression level of *TAA1a* in the MS line was significantly lower than that in the MF line in the four periods assessed (Figure 2B).

The amino acid sequences of TaTDRL and TaMYB103 were also analyzed and compared with those of their putative rice (OsTDR and OsMYB103, respectively) and *Arabidopsis* (AtAMS and AtMYB103, respectively) homologs. The sequence similarity between TaTDRL and OsTDR was 57.8%, and that between TaTDRL and AtAMS was 27.58%. All three proteins possessed a typical bHLH domain, indicating that TaTDRL homologs are highly conserved in different species (Appendix A). The sequence similarity between TaMYB103 and OsMYB103 was 79.89%, and that between TaMYB103 and AtMYB103 was 53.82%. They all possessed a typical R2R3-MYB domain and a highly similar N-terminal DNA-binding domain (Appendix A). A homology analysis showed that TaTDRL has a close evolutionary relationship with *Triticum urartu* (EMS48440.1) and *Hordeum vulgare* (KAE8787147.1), while TaMYB103 has a close evolutionary relationship with *Hordeum vulgare* (KAE8813946.1) (Appendix A).

Additionally, TaTDRL and TaMYB103 were fused with EGFP for subcellular localization analysis in tobacco leaves. Whereas the EGFP signal of the empty vector could be detected in the entire cell, the green fluorescence signals of TaTDRL-EGFP and TaMYB103-EGFP were limited to one location in the cell, likely the nucleus. To confirm this possibility, we fused the reported nuclear marker gene *OsSK2* with ERFP (OsSK2-ERFP) and co-expressed it with TaTDRL-EGFP or TaMYB103-EGFP in tobacco leaves, and found that green and red fluorescence signals overlapped (Appendix A), indicating that TaTDRL and TaMYB103 are both nuclear-localized proteins. Xia et al. previously reported that TaTDRL has no toxic effect on the growth of yeast, and that its N-terminal region (aa 1–294) is required for its transcriptional activation activity [46] (Appendix A). Similarly, we found that TaMYB103 was not toxic to the growth of yeast and also displayed transcriptional activation activity (Appendix A). These observations indicated that both TaTDRL (bHLH family) and TaMYB103 (R2R3 MYB family) are likely to function as nuclear-localized transcription factors. The tissue expression patterns of *TaTDRL* and *TaMYB103* showed similar trends. Both were highly expressed in young panicles and anthers, but their expression levels were relatively low in roots, stems, and leaves (Figure 2C,E). Next, we compared the relative expression levels of both genes during different stages of anther development between the MF and MS lines (Figure 2D,F). We found that in the MF line, the expression levels of *TaTDRL* and *TaMYB103* gradually decreased with microspore development, with the highest expression being observed at the early uninucleate stage. However, in the MS line, *TaTDRL* expression first increased and then decreased with microspore development, which was similar to the trend seen for *TAA1a*; however, the expression level of *TaMYB103* was low during the four periods assessed. The expression levels of these two genes in the early uninucleate stage were significantly lower in the MS line than in the MF line. These results suggested that the differences in *TaTDRL* and *TaMYB103* expression levels between the MS and MF lines in the early uninucleate stage may be strongly associated with the differences observed in anther or pollen development in wheat, and their interactions and regulatory patterns with *TAA1a* need further study.

### 2.4. TaTDRL and TaMYB103 Synergistically Activate the Expression of TAA1a

Fragments of TaTDRL without self-activation activity (1–344 aa, 295–344 aa, 295–557 aa, and 345–557 aa) were respectively inserted into the pGBKT7 yeast expression vector (Figure 3A shows only the 345–557 aa segment). The coding sequence (CDS) of TaMYB103 was inserted into pGADT7, yielding TaMYB103-AD. The results of the yeast two-hybrid assay showed that the yeast strains harboring TaTDRL∆C-BD (345–557 aa) and TaMYB103-AD grew normally on medium containing a quadruple dropout supplement, and in the X-α-Gal assay, the colonies turned blue (Figure 3A). The interaction between the TaTDRL and TaMYB103 proteins was validated using bimolecular fluorescence complementation (BiFC). The results showed that TaTDRL-nYFP and TaMYB103-cYFP expression overlapped and the YFP signals were detected in nuclei (Figure 3B). Thus, both the yeast two-hybrid assay and BiFC experiments indicated that the TaTDRL and TaMYB103 proteins interacted and that TaMYB103 interacts with the C-terminal of TaTDRL (345–557 aa).

The interaction between TaTDRL and TaMYB103 with the corresponding *cis*-acting elements (E-box and CCAACC) on the *TAA1a* promoter was verified by a yeast one-hybrid assay. We found that yeast cells co-transformed with the corresponding section of *proTAA1a* and TaTDRL-AD or TaMYB103-AD, but not pGADT7 (empty vector), could grow on SD/−Leu/+AbA (200 ng/mL) (Figure 4B). These findings indicated that TaTDRL and TaMYB103 can bind to *proTAA1a* through the CACGTG and CCAACC motifs, respectively. The dual-luciferase reporter system was then used to test whether TaTDRL and TaMYB103 exert transcriptional effects on *proTAA1a* (the constructs used in the dual-luciferase reporter assay are shown in Figure 4A). Compared with the negative control (*p35S::nos* + *proTAA1a::LUC*), the LUC/REN ratio was increased by 3.11- and 4.52-fold, respectively, in *Nicotiana benthamiana* leaves co-transformed with *p35S::TaTDRL* + *proTAA1a::LUC* or *p35S::TaMYB103* + *proTAA1a::LUC*. Notably, when *p35S::TaTDRL* and *p35S::TaMYB103* were simultaneously co-transformed into tobacco leaves with *proTAA1a::LUC*, the LUC/REN ratio was 7.38-fold that of the negative control (Figure 4A,C). These results showed that TaTDRL and TaMYB103 can directly regulate and synergistically activate the expression of *TAA1a* in wheat.

### 2.5. Pollen Development and the Expression of MS2 Were Dysregulated in TaTDRL-EAR and TaMYB103-EAR Plants

We further analyzed the role of *TaTDRL* and *TaMYB103* in anther and pollen development using *A*. *thaliana* transgenic lines overexpressing *TaTDRL* or *TaMYB103* (TaTDRL-OE and TaMYB103-OE, respectively) as well as transgenic lines expressing chimeric repressors of the two transcription factors (TaTDRL-EAR and TaMYB103-EAR, respectively). *A*. *thaliana* seedlings were screened using hygromycin B. To exclude false positives, a qRT-PCR was used to detect positive plants from among the T1 generation. According to the relative expression levels of transcription factors, the four recombinant plasmids were successfully transferred into *Arabidopsis* (Appendix A). For T1 transgenic plants, a single plant was harvested. In subsequent generations, hygromycin B was used to screen the transgenic lines until pure lines were obtained. As shown in Figure 5A–E, no differences in floret blooming were observed between the transgenic *Arabidopsis* plants and WT plants. The fertility of mature pollen was further analyzed using Alexander staining. The results showed that the pollen in WT anthers stained red and almost all the pollen grains were alive (Figure 5F). In the anthers of the TaTDRL-OE and TaMYB103-OE transgenic lines, the pollen also stained red, and indicating that most of them were active pollens (Figure 5G,I). In contrast, the pollen in the anthers of the TaTDRL-EAR and TaMYB103-EAR transgenic lines was bluish and mostly lifeless (Figure 5H,J). Quantitative analysis after staining indicated that the percentages of viable pollen grains in the TaTDRL-EAR and TaMYB103-EAR lines were 44.82% (Figure 5M) and 54.17% (Figure 5O), respectively, significantly lower than that in the WT plants (Figure 5K) or those in the overexpression lines (Figure 5L,N).

SEM was used to analyze the outer epidermal cells and pollen grains of transgenic and WT plants. No significant differences in anther epidermal structure were detected between WT and TaTDRL-EAR plants. However, the anther epidermis in the TaTDRL-EAR line showed some deformity compared with the WT (Figure 6A,E). The pollen of WT plants in the binucleate stage was full and spherical, and the pollen exine had formed (Figure 6B). In contrast, many of the pollen grains in the TaTDRL-EAR line had withered and collapsed (Figure 6F), whereas those in the TaMYB103-EAR line appeared to be normal at this stage (Figure 6I). In the late trinucleate stage, WT pollen matured and formed normally, and the pollen grain surface presented an extended network structure (Figure 6C,D). However, the pollen grains of the TaTDRL-EAR and TaMYB103-EAR lines had shriveled heteromorphism (Figure 6G,J,K), and the network structure on their surfaces was also shriveled and depressed (Figure 6H,L).

Siliques of TaTDRL-OE, TaMYB103-OE, TaTDRL-EAR and TaMYB103-EAR lines were analyzed. Compared with WT *Arabidopsis* and overexpressed *Arabidopsis* lines, the pods of the EAR-expressing lines were significantly shorter. Additionally, the ovules of the overexpression lines were better developed than those of the WT, and there was almost no shriveled endosperm; conversely, numerous unfertilized and shriveled ovules could be seen in the fruit pods of both the TaTDR-EAR and TaMYBB103-EAR lines (Figure 6M,O). The seed setting rate of the three lines was statistically quantified, and it was found that the seed setting rate of the TaTDR-EAR line was only about 50%, and that of the TaMYBB103-EAR line was only about 40%; (Figure 6N,P). In contrast, the seed setting rates of the overexpression lines were significantly higher than that of the WT. The expression levels of *AtMS2* were also measured both in transgenic and WT plants. The expression level of *MS2* in the overexpression lines was significantly higher than that in the WT line at stages 9 and 10, whereas that in the EAR-expressing lines was significantly lower (Figure 7A,B). These results suggested that the transcription factors TaTDRL and TaMYB103 are crucial positive regulators of pollen development.

## 3. Discussion

Previous studies have shown that the male sterile line with *Aegilops kotschyi* cytoplasm in wheat have the characteristics of being stable in sterility, complete sterility and the restoration resource are widely, and can be used for the study of male sterility and heterosis in wheat [47,48,49]. MS(KOTS)-90-110 is a typical male-sterile line with *Aegilops kotschyi* cytoplasm in wheat. To study the physiological basis of and molecular mechanism underlying pollen abortion in the MS(KOTS)-90-110 line, we previously created its near-isogenic lines that differ only in terms of fertility [43]. Because common wheat is a heterohexaploid crop with a large (40 times the size of rice and 5.5 times that of humans) and complex genome, the development of gene cloning and molecular design breeding techniques for important agronomic traits lags far behind that of rice and maize [50]. Accordingly, it is of great theoretical importance to identify the key genes involved in the regulation of wheat fertility and explore the molecular mechanism associated with pollen abortion in a male-sterile line with the *Aegilops kotschyi* cytoplasm, which would allow the improved utilization of wheat heterosis.

Sporopollenin is the main component of pollen exine and its biosynthesis is closely related to lipid metabolism. In the sporopollenin synthesis pathway in *Arabidopsis*, acetyl-CoA produced via the tricarboxylic acid cycle is first modified and transported to the endoplasmic reticulum by Acyl-CoA Synthetase 5 (ACOS5), encoded by the *ACOS5* gene, which is specifically expressed in the tapetum [28]. Subsequently, the modified lipids are hydroxylated into fatty acids by CYP703A2 and CYP704B1 [23,51], and then these hydroxylated fatty acids are converted to fatty alcohols by the fatty acid reductase (*Male sterility 2*, *MS2*) [30]. Finally, these fatty alcohols are transported from the tapetum by ATP binding cassette transporter G26 (ABCG26) to the microspore surface for polymerization, after which they are deposited on primexine, and eventually form sporopollenin [52,53]. These observations stress the importance of investigating the function of genes involved in lipid metabolism in the tapetum to better understand the mechanisms involved in male sterility. Relevant studies have shown that the *DPW* gene in rice (homologous to *MS2*) regulates the formation of sporopollen and Ubisch bodies and participates in the development of the outer wall of rice pollen [54]. *TAA1a* is a wheat homolog of *MS2*, and has been proposed to function as a fatty acyl reductase with a role in the synthesis of pollen exine [34]. In this study, qRT-PCR analysis showed that *TAA1a* expression was highest in the anthers. Moreover, the expression level of *TAA1a* was significantly lower in the MS line than in the MF line, and the expression level of *TAA1a* in the MF line was highest in the late uninucleate stage (Figure 2A,B). Our results further showed that two transcription factor-encoding genes (*TaTDRL* and *TaMYB103*) isolated from the MS line are important modulators of wheat anther development, likely through co-regulating wheat microspore development via the co-activation of *TAA1a* promoter activity. TaTDRL contains a typical bHLH domain and functions as a nuclear transcription factor [46]. In this study, we found that TaMYB103 is a R2R3-MYB transcription factor that also localizes to the nucleus. A homology analysis indicated that *TaTDRL* and *TaMYB103* are homologs of *AtAMS* and *AtMYB103* (*MS188*) of *A*. *thaliana*, respectively. Both *TaTDRL* and *TaMYB103* were found to be widely expressed in many wheat tissues and organs, including roots, stems, leaves, anthers, and young panicles. However, the highest expression levels of these genes were detected in anthers (Figure 2C,E), indicating that they may play an important role in anther or pollen development. In *Arabidopsis*, mutations in *AtAMS* and *AtMYB103* result in abnormal tapetal degradation [35]. Ying et al. [55] showed that the tapeta of MS(KOTS)-90-110 plants were completely degraded at the late uninucleate stage. Moreover, abnormal tapetum degradation is reported to be the key factor leading to microspore abortion [36,56]. In our study, the microspores of MS(KOTS)-90-110 plants exhibited plasmolysis in the late uninucleate stage, abnormal morphology in the binuclear and trinuclear stages, and the mature pollen grains lacked starch accumulation (Figure 1A–L). These abortion-related characteristics may be associated with the dysregulation of tapetal gene expression. The relative expression levels of *TaTDRL* and *TaMYB103* in the early uninucleate stage were significantly higher in the fertile near-isogenic lines than in the male-sterile lines (Figure 2D,F). Combined with the above, the relative expression of *TAA1a*, *TaTDRL*, and *TaMYB103* in wheat anthers showed the same trend. Therefore, it is speculated that there is some interaction between the three, so as to regulate the anther development of the wheat MS(KOTS)-90-110 line.

In *A*. *thaliana*, *MYB33* and *MYB65* play important roles in tapetal development. In *myb33* and *myb65* double mutants, tapetal cell hypertrophy during meiosis leads to pollen abortion [57]. Another MYB gene, *MYB32*, participates in the phenylpropanoid metabolic pathway and regulates the formation of pollen exine in *A*. *thaliana* [58]. Meanwhile, *DYT1* encodes a bHLH transcription factor that is expressed during early anther development. This gene regulates the callose degradation pathway in the tapetum and its dysregulation affects microspore development. The abnormal vacuolation of tapetal cells in *dyt1* mutants leads to premature microspore degradation [59,60]. These observations highlight the important roles of MYB and bHLH family proteins in pollen development. Several studies have shown that rice TDR, also a bHLH transcription factor, plays an important role in regulating tapetal development and pollen formation [61,62,63,64]. Rice *OsMTB103-RNAi* lines display developmental defects in the tapetum, and the pollen of related transgenic lines is sterile [65]. Studies have shown that MYB and bHLH proteins interact and synergistically regulate multiple biochemical pathways. For example, *GL3* and *TRANSPARENT TESTA8*, members of the bHLH family, and *TRANSPARENT TESTA2*, *GL1*, and *MYB61*, all MYB family members, synergically regulate the anthocyanin synthesis pathway [66,67]. bHLH-MYB complexes control jasmonate-mediated stamen and seed maturation [68]. In tapetal cells, AMS and MS188 regulate the biosynthesis of sporopollenin by binding to the *CYP703A2* promoter [39]. In *Arabidopsis*, meanwhile, MS188 is a key regulator of genes involved in activating the sporopollenin synthesis pathway; the *AMS* gene, which encodes a bHLH transcription factor, is an important co-factor for MS188 in this process. The two synergistically regulate the expression of key genes related to sporopollenin synthesis, such as *CYP703A2*, *CYP704B1*, *PKSA*, *PKSB*, *ACOS5*, and *TKPR1* [40]. This is similar to that reported for the MYB family transcription factor TDF1, which can directly regulate *AMS* expression, as well as interact with AMS to regulate the expression of downstream genes via a feed-forward loop [69]. In this study, using yeast two-hybrid and BiFC assays, we demonstrated that TaTDRL and TaMYB103 directly interact (Figure 3A,B). Then, using a yeast one-hybrid assay, we further demonstrated that TaTDRL and TaMYB103 can bind to specific motifs on *proTAA1a* (Figure 4B). Importantly, dual-luciferase assay results demonstrated that both TaTDRL and TaMYB103 can independently activate transcription from the *TAA1a* promoter (*proTAA1a*), and the activity of *proTAA1a* was noticeably stronger when TaTDRL and TaMYB103 were co-expressed than when each was expressed alone (Figure 4C). We have previously reported that MS(KOTs)-90-110 pollen abortion is characterized by early tapetum degradation during anther development. Importantly, compared with normal fertile microspores, sporopollenin accumulation was much lower in microspores of the in MS(KOTs)-90-110 line, which significantly hindered pollen exine formation [55]. Using transgenic *Arabidopsis* plants, we showed that, when TaTDRL or TaMYB103 functioned as repressors, the pollen wall was shrunken and depressed (Figure 6H,L), possibly due to the failure of sporopollenin synthesis, and a similar phenotype was observed in the microspores of TaTDRL-EAR and TaMYB103-EAR plants (Figure 6G,J,K). Additionally, when we treated the anthers of WT and transgenic plants with Alexandria dye solution, a large number of inactive pollen grains could be seen in the anthers of TaTDRL-EAR and TaMYB103-EAR plants (Figure 5H,J). We further found that the seed setting rate of *Arabidopsis* fruit pods in TaTDRL-EAR and TaMYB103-EAR transgenic plants was significantly lower than that in WT plants (Figure 6N,P). These results indicated that TaTDRL and TaMYB103 exert a positive regulatory effect on microspore development in transgenic *Arabidopsis*. To further explore whether the microspore defects were related to sporopollenin synthesis, we assessed the relative expression level of *AtMS2* in the anthers, and found that the *AtMS2* expression level was significantly higher in the overexpression lines than in WT plants, whereas the opposite was observed in the TaTDRL-EAR and TaMYB103-EAR lines (Figure 7).

Combined, these findings suggest that TaTDRL, a bHLH family member, and TaMYB103, a MYB family protein, may be implicated in a feed-forward regulatory mechanism during anther development in wheat, similar to that reported for AtAMS and AtMYB103 in *Arabidopsis*, and thus influence pollen fertility by regulating *TAA1a* transcription in the sporopollenin synthesis pathway. This possible transcriptional regulatory mechanism is illustrated in Figure 8. Because both *AtAMS* and *AtMYB103* are key regulators of tapetal development in *Arabidopsis*, it is important to further elucidate the potential regulatory functions of *TaTDRL* and *TaMYB103* in wheat tapetal development. Similarly, the roles of *TaTDRL* and *TaMYB103* in wheat male sterility, and the transcriptional regulation of *TAA1a* promoter activity in this process, also need to be addressed using transgenic wheat.

## 4. Materials and Methods

### 4.1. Plant Materials and Growth Conditions

In this study, a pair of near-isogenic lines of K-type CMS wheat—MS(KOTS)-90-110 (MS line) and MR(KOTS)-90-110 (MF line)—was used as the research material. Seeds were sown in an experimental field of Northwest A&F University, Yangling, Shaanxi, China (108°82′ E, 34°15′ N) from 2017 to 2019 (as usual). Ten anthers were collected and mixed at each of the early uninucleate stage, late uninucleate stage, binucleate stage, and trinucleate stage of microspore development. All samples were immediately frozen in liquid nitrogen and stored at −80 °C for further analysis. All experiments were repeated three times.

*A*. *thaliana* ecotype Columbia (Col-0) was derived from our laboratory. The seeds were surface-sterilized in a 10% NaClO solution for 8 min and washed five times with sterilized water. Seeds were sown on 1/2 Murashige and Skoog (MS) medium (0.9% [*w*/*v*] agar, 1% [*w*/*v*] sucrose, pH 5.9]. After three days of stratification at 4 °C, the plates were transferred to a plant growth incubator for seven days before being transferred to soil and grown under a 16-h light/8-h dark photoperiod at 22 °C in a growth room.

### 4.2. Isolation of the TAA1a Promoter and Bioinformatics Analysis

A pair of primers for the amplification of the *TAA1a* promoter sequence (*proTAA1a*) (*TAA1a* bproF: 5′-CTAGTCTCTTACTACTGAGTCCCC-3′ and *TAA1a* bproR: 5′-AACCATCTTGATTCTTTTCTTG-3′) was designed based on the published *TAA1a* sequence (GenBank: AJ459250.1). A 1315-bp (−1309 bp to +6 bp; the adenosine residue of the ATG initiation codon was numbered as +1) region of the *TAA1a* promoter was amplified from the genomic DNA of the MR(KOTS)-90-110 line and cloned into the pEASY-Blunt cloning vector. Fragment insertion was confirmed by sequencing.

The *cis*-acting elements of the full-length *TAA1as* promoter sequence (−1309 to −1 bp) were identified using the online programs PLACE (http://www.dna.afrc.go.jp/PLACE/signalup.html, accessed on 15 June 2020) [70] and PlantCARE (http://bioinformatics.psb.ugent.be//webtools/plantcare/html, accessed on 27 June 2020) [71].

### 4.3. Expression Analysis

The Integrated DNA Technologies website (https://sg.idtdna.com/scitools/Applications/RealTimePCR/Default.aspx, accessed on 27 September 2020) was used for the design of primers targeting gene-specific regions. The relative transcript levels of wheat and *Arabidopsis* genes were normalized to those of the internal controls *TaActin* (GenBank: AB181991.1) and *EF1a* [72], respectively (Appendix A). Total RNA was extracted using TRIzol reagent (Takara Bio, Tokyo, Japan) and served as the template for reverse transcription, which was carried out with the Easy Script One-step gDNA Removal and cDNA Synthesis SuperMix (TransGen Biotech, Beijing, China). Synthesized cDNA (200 ng per gene) was used as the template for quantitative real-time PCR (qPCR). Primers were designed using Primer Premier 5.0 software and are listed in Appendix A. Relative gene expression levels were calculated using the 2^−ΔΔCt^ method. Each experiment was performed in three independent replicates [73].

### 4.4. Transformation of Arabidopsis thaliana

Studies have shown that AMS can bind the CANNTG 6-bp motif in vitro in *Arabidopsis* and that MYB103 preferentially binds sequences containing the CCAACC motif [35,74]. Bioinformatics analysis of *proTAA1a* showed that the CANNTG (N = A/G/C/T) element existed at −1269 bp, −1252 bp, −1063 bp, −590 bp, −429 bp, −331 bp, and −159 bp, while a CCAACC element could be found at −1210 bp (Appendix A). A homology analysis indicated that *TaTDRL* and *TaMYB103* were the wheat homologs of *AMS* and *MYB103*, respectively, which encode two transcription factors. Accordingly, they were used to further investigate the transcriptional regulation of *TAA1a* during pollen development. The full-length CDSs of *TaTDRL* and *TaMYB103* were amplified from mixed MR(KOTS)-90-110 anther cDNA using KOD DNA Polymerase (Toyobo; https://www.toyobo-global.com/, 27 December 2020). Primers were designed using Primer Premier 5.0 software and are listed in Appendix A.

All fragments were sub-cloned into the pCAMBIA1302 vector between the NcoI and SpeI restriction sites. Vectors for the overexpression of TaTDRL and TaMYB103 were constructed by inserting the respective sequences downstream of the 35S promoter, yielding *p35S::TaTDRL* (TaTDRL-OE) and *p35S::TaMYB103* (TaMYB103-OE). To construct a chimeric repressor for each of the transcription factors, the CDSs of *TaTDRL* and *TaMYB103* were respectively fused with DNA encoding the EAR-motif repressor domain (RD) (LDLDLELRLGFA). The resulting constructs were subsequently inserted downstream of the 35S promoter of the pCAMBIA1302 vector, generating *p35S::TaTDRLRD* (TaTDRL-EAR) and *p35S::TaMYB103RD* (TaMYB103-EAR). Details of the primers used for generating these constructs are listed in Appendix A.

*A*. *thaliana* ecotype Columbia (Col-0) plants were grown in a growth room of a phytotron for five to six weeks under controlled conditions (16-h light/8-h dark photoperiod; temperature: 22 °C). Recombinant plasmids were introduced into the *Agrobacterium tumefaciens* strain GV3101 and then transformed into *Arabidopsis* using the floral dip method [75]. The transformed *Arabidopsis* plants were placed in a horizontal position in the dark for 24 h and then allowed to grow until seeds were obtained from the T0 generation. Transgenic lines were grown on 1/2 MS medium supplemented with 50 mg/L hygromycin B. Transcript analysis was performed for the T2 transgenic *Arabidopsis* lines using qRT-PCR. Representative homozygous T3 progeny were selected for further studies based on transcript analysis [46].

### 4.5. Analysis of Protein Subcellular Localization

The *TaTDRL* and *TaMYB103* CDSs were fused to the N-terminus of enhanced GFP (EGFP) in the pCAMBIA1302-2×MYC-EGFP vector, yielding the recombinant plasmids (TaTDRL-EGFP and TaMYB103-EGFP, under the control of the *CaMV35S* promoter) used in subcellular localization experiments. The primers used for plasmid construction are listed in Appendix A. To show that the TaTDRL-EGFP and TaMYB103-EGFP fusion proteins were localized to the nucleus, the nuclear marker gene *OsSK2* [76] fused with ERFP (OsSK2-ERFP) was co-expressed with TaTDRL-EGFP and TaMYB103-EGFP in tobacco leaf epidermal cells. The recombinant plasmids were transformed into *A*. *tumefaciens* GV3101 using a freeze–thaw method. EGFP fluorescence was observed and imaged using laser confocal microscopy [77].

### 4.6. Yeast One-Hybrid Assay

Analysis of the *TAA1a* promoter sequence revealed the presence of putative TaTDRL-binding and TaMYB103-binding motifs (CANNTG [N = A/T/C/G ] and CCAACC, respectively). To test whether TaTDRL and TaMYB103 can bind to these two motifs, TaTDRL and TaMYB103 cDNAs were respectively cloned into the pGADT7 vector by homologous recombination (primer details are provided in Appendix A). Tandem repeats of the respective sections of the *TAA1a* promoter (TaTDRL-promoter and TaMYB103-promoter) were respectively inserted into the pAbAi vector, yielding bait vectors (TaTDRL-promoter-pAbAi and TaMYB103-promoter-pAbAi). The vectors were then transformed into the Y1HGold yeast strain and cultured on SD/−Leu/+AbA medium (200 ng/mL AbA).

### 4.7. Yeast Two-Hybrid Assay

To detect the toxicity and self-activation activity in yeast, seven expression vectors were constructed, namely: TaTDRL-BD (1–557 aa), TaTDRL-NBD (1–294 aa), TaTDRL-HLHBD (295–344 aa), TaTDRL-CBD (345-557 aa), TaTDRL-NHLHBD (1–344 aa), TaTDRL-HLHCBD (295–557 aa), and TaMYB103-NBD (1–352 aa) (primer details are provided in Appendix A). All the plasmids were cultured in SD/−Trp liquid medium. Their self-activation activity was detected on SD/−Trp/−His/−Ade solid medium. The toxicity of the target protein was determined by comparing the growth in culture medium of Y2H cells transformed with pGBKT7 and that of cells transformed with the recombinant plasmids.

For interaction analysis, the appropriate plasmids were co-transformed into the Y2HGold yeast strain. The transformed cells were coated on SD/−Trp/−Leu medium and cultured at 30 °C for two to four days. The interaction between the combinations was tested on medium containing a quadruple dropout supplement (SD/−Trp/−Leu/−His/−Ade). X-α-Gal was used to verify positive interactions.

### 4.8. BiFC

For the BiFC assay, the full-length genes (minus the stop codons) of *TaTDRL* and *TaMYB103* were amplified by PCR using primers specific to *TaTDRL* and *TaMYB103* and containing the BstBI restriction site. Primer details are provided in Appendix A. TaTDRL-nYFP and TaMYB103-cYFP recombinant vectors were obtained by fusing the respective cDNA sequences with the N-terminal and C-terminal portions of YFP in the BiFC vector. Bacterial solutions containing the TaTDRL-nYFP and TaMYB103-cYFP recombinant plasmids were mixed in equal volumes, allowed to stand at room temperature for 1 h, and then injected into *N*. *benthamiana* leaves with a syringe. The YFP fluorescence signal was observed using a laser confocal microscope [77,78].

### 4.9. Dual-Luciferase Assay

Dual-luciferase assays are widely used to evaluate the regulatory effects of transcription factors on target promoters [79,80]. The dual-reporter vector pGreenII 0800-LUC includes a native firefly luciferase (LUC) and a *CaMV35S* promoter-driven, internally controlled *Renilla* luciferase (REN) [81].

*proTAA1a* was cloned into the pGreenII 0800-LUC reporter vector, yielding *proTAA1a::LUC*. The transcriptional activity of TaTDRL and TaMYB103 was assayed by inserting the complete CDSs of *TaTDRL* and *TaMYB103* into the pGreenII 62-SK effector plasmid to generate *p35S::TaTDRL* and *p35S::TaMYB103* (the primers used are listed in Appendix A). Three experiments were designed. *proTAA1a::LUC* + *P35S::TaTDRL*, *proTAA1a::LUC* + *P35S::TaMYB103*, or *proTAA1a::LUC* + *P35S::TaTDRL* + *P35S::TaMYB103* were co-transformed into *A*. *tumefaciens* GV3101 (pSoup-p19). *A*. *tumefaciens* transfected with *proTAA1a::LUC* + *p35S::nos* served as negative control. All combinations consisted of a 1:1 volume ratio and were injected into *N*. *benthamiana* leaves with a syringe. The dual-luciferase assay (Promega, Madison, WI, USA) was performed 72 h after transfection. Absolute LUC and REN activity levels were determined using a GLOMAX 96 microplate photometer (Promega, Madison, WI, USA). At least three biological replicates were performed for each combination.

### 4.10. Phenotypic Characterization and Microspore Analysis of Wheat and Arabidopsis thaliana

Wheat anthers were observed under a Motic K400 dissecting microscope (Preiser Scientific, Louisville, KY, USA) and photographed using a Nikon E995 digital camera (Nikon, Tokyo, Japan). The stages of microspore development were identified by staining the anthers with 1% acetocarmine and 2% I_2_-KI. Samples were photographed using a DS-U2 high-resolution camera mounted on a Nikon ECLIPSE E600 fluorescence microscope (Nikon, ECLIPSE, E600, Tokyo, Japan) and analyzed using NIS-Elements software (Nikon, Tokyo, Japan) [12,82].

The prepared Alexander staining solution was divided into eight tubes. The *A*. *thaliana* buds were completely immersed in the dye solution and placed in the dark at room temperature for 3 h. The samples were then observed and photographed using a fluorescence microscope. Formula reference for dyeing solution [83]. Before reaching maturity (approximately 35 days), *Arabidopsis* fruit pods were opened using an anatomy needle, and the seed setting rates of the OE and EAR lines were determined with a stereoscopic microscope (OLYMPUS, Japan). WT fruit pods served as control. At least three biological replicates were performed for each line. SEM was used to determine the surface characteristics of the anthers. For this, anthers in the 12th and 13th stage of development were fixed in 4% glutaraldehyde, treated with an alcohol gradient, dried, and broken in sequence. Finally, the anthers and pollen grains were mounted on a stub with colloidal silver and imaged using a JSM-6360LV scanning electron microscope (JEOL, Tokyo, Japan) [84].

## 5. Conclusions

In this study, two transcription factor-encoding genes (*TaTDRL* and *TaMYB103*) and the promoter of their direct target gene (*proTAA1a*) were identified and isolated from common wheat. The transcriptional regulatory mechanism was found to involve the synergistic activation of the activity of *proTAA1a* by TaTDRL and TaMYB103. CRES-T was used to study the role of *TaTDRL* and *TaMYB103* in microspore development in *Arabidopsis*. The results showed that the TaTDRL-EAR and TaMYB103-EAR lines had abnormal microspore development and a significantly lower seed setting rate compared with the WT line and the corresponding overexpression lines. In summary, we identified a potential transcriptional regulatory mechanism associated with pollen development in wheat and provided a theoretical molecular basis for further revealing the mechanisms underlying CMS anther development.

## Figures and Tables

**Figure 1 ijms-23-07996-f001:**
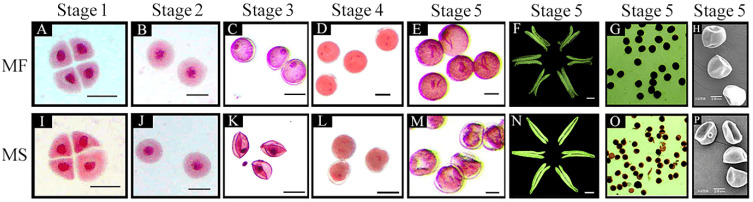
Phenotypes of anthers and microspores development in the male-sterile (**MS**) line and its near-isogenic male fertile (**MF**) line. Stage 1: tetrad stage (**A**,**I**); stage 2: early uninucleate stage (**B**,**J**); stage 3: late uninucleate stage (**C**,**K**); stage 4: binucleate stage (**D**,**L**); stage 5: trinucleate stage (**E**–**H**,**M**–**P**). Scale bars: 50 µm in (**A**–**E**,**I**–**M**), 500 µm in (**F**,**N**), 20 µm in (**H**,**P**).

**Figure 2 ijms-23-07996-f002:**
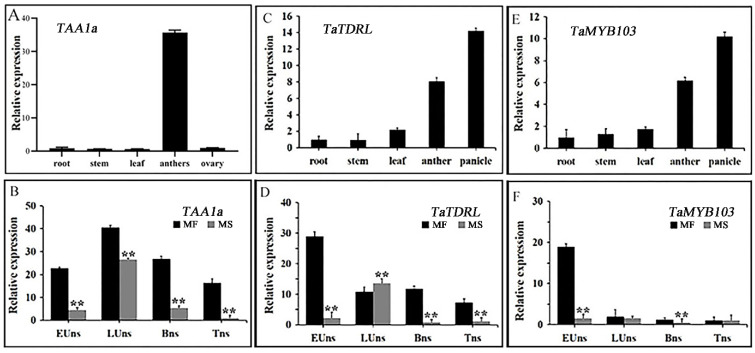
The expression patterns of *TAA1a, TaTDRL*, and *TaMYB103*. (**A**,**C**,**E**) The expression pattern of *TAA1a, TaTDRL*, and *TaMYB103* in various tissues in the male-sterile (MS) line and its near-isogenic male-fertile (MF) line. (**B**,**D**,**F**) The expression levels of *TAA1a*, *TaTDRL*, and *TaMYB103* in the MF and MS lines at different stages of anther development. EUns: Early uninucleate stage; LUns: Late uninucleate stage; Bns: Binucleate stage; Tns: Trinucleate stage. Data are presented as the means ± SD of three technical replicates and three independent biological replicates. The significance of differences was assessed using the Student’s *t*-test. ** *p* < 0.01.

**Figure 3 ijms-23-07996-f003:**
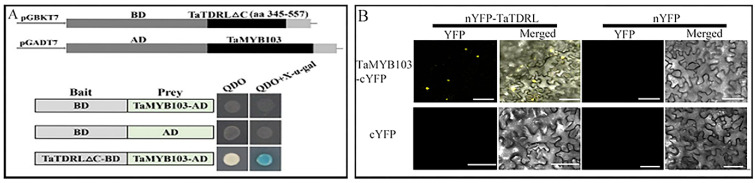
The interaction between TaTDRL with TaMYB103. (**A**) Yeast two-hybrid assay to test the interaction of TaTDRL with TaMYB103. QDO, SD/−Trp/−Leu/−His/−Ade medium. (**B**) Bimolecular fluorescence complementation (BiFC) assay for confirmation of the interaction between TaTDRL and TaMYB103. Scale bars: 50 μm in (**B**).

**Figure 4 ijms-23-07996-f004:**
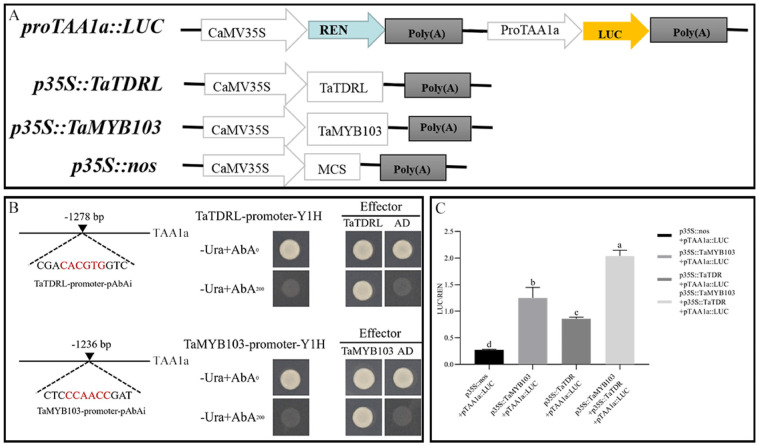
Co-regulation of *proTAA1a* by TaTDRL and TaMYB103. (**A**) Constructs used in the dual-luciferase assay. (**B**) Analysis of the interaction between TaTDRL or TaMYB103 and the *TAA1a* promoter using a yeast one-hybrid assay. (**C**) Transient dual-luciferase assays were performed in the *Nicotiana benthamiana* leaf. The activities of firefly luciferase (LUC) and *Renilla* luciferase (REN) were sequentially measured and the LUC/REN ratio was calculated as the final transcriptional activation activity. Data represent the means ± standard deviation of three biological replicates. Different lower-case letters indicate significant differences at *p* < 0.05.

**Figure 5 ijms-23-07996-f005:**
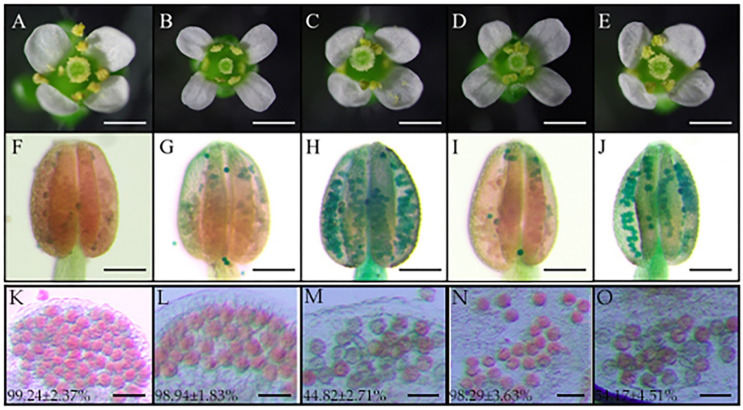
Phenotypic analyses of wild-type, TaTDRL-OE, TaTDRL-EAR, TaMYB103-OE, and TaMYB103-EAR transgenic *Arabidopsis* plants. (**A**–**E**) Phenotypic analysis of flowers; (**F**–**O**) Alexander staining of a representative anther; (**A**,**F**,**K**) wild-type; (**B**,**G**,**L**) TaTDRL-OE; (**C**,**H**,**M**) TaTDRL-EAR; (**D**,**I**,**N**) TaMYB103-OE; (**E**,**J**,**O**) TaMYB103-EAR. Scale bars: 2000 µm in (**A**–**J**), 50 µm in (**K**–**O**). Data are presented as means ± SD of three technical replicates and three independent biological replicates.

**Figure 6 ijms-23-07996-f006:**
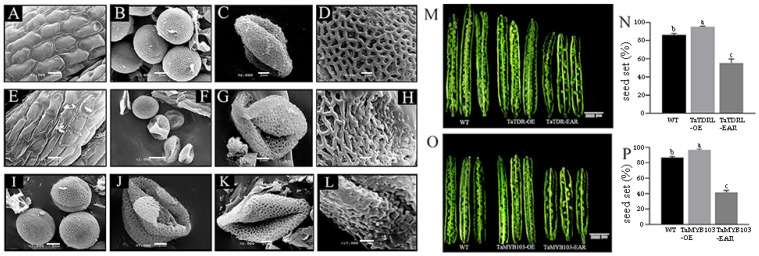
Scanning electric microscopic (SEM) evaluation of the microspores and analysis of the fruit pods of transgenic *Arabidopsis*. (**A**–**D**) Microspores of wild-type (WT) *Arabidopsis*. (**E**–**H**) Microspores of the TaTDRL-EAR line. (**I**–**L**) Microspore of the TaMYB103-EAR line. (**A**,**E**) Outer anther epidermis. (**B**,**F**,**I**) Microspores in the binucleate stage. (**C**,**G**,**J**,**K**) Microspores in the trinucleate stage. (**D**,**H**,**L**) Microspore network structure. (**M**) Fruit pods of the WT, TaTDRL-OE, and TaTDRL-EAR lines. (**O**) Fruit pods of the WT, TaMYB103-OE, and TaMYB103-EAR lines. (**N**) The seed setting rate of the WT, TaTDRL-OE, and TaTDRL-EAR lines. (**P**) The seed setting rate of the WT, TaMYB103-OE, and TaMYB103-EAR lines. Scale bars: 10 µm in (**A**,**E**,**F**), 5 µm in (**B**,**I**), 2 µm in (**C**,**G**,**J**,**K**), 1 µm in (**D**,**H**,**L**), 2000 µm in (**M**,**O**). Data represent the means ± standard deviation of three biological replicates. Different lower-case letters indicated significant differences at *p* < 0.05.

**Figure 7 ijms-23-07996-f007:**
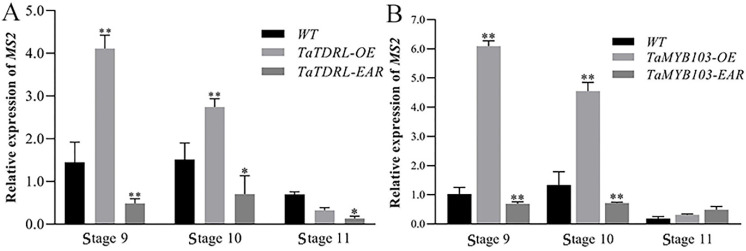
The expression levels of the *AtMS2* gene in transgenic *Arabidopsis* and wild-type plants. (**A**) The expression levels of *AtMS2* in wild-type (WT), TaTDRL-OE, and TaTDRL-EAR plants. (**B**) The expression levels of *AtMS2* in WT, TaMYB103-OE, and TaMYB103-EAR plants. Stage 9 to Stage 11: Anther development stages in *Arabidopsis*. The significance of differences was assessed using the Student’s *t*-test. * *p* < 0.05, ** *p* < 0.01.

**Figure 8 ijms-23-07996-f008:**
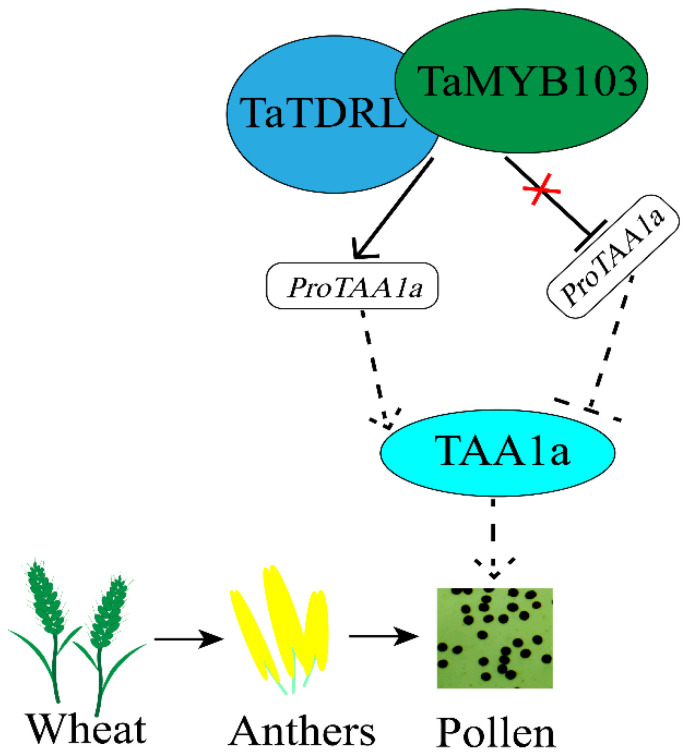
A potential model in which TaTDRL and TaMYB103 synergistically activate *ProTAA1a* to regulate wheat pollen development.

## Data Availability

The data presented in this study are available in the article and Appendix A.

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
