# Peer review of "The Transcription Factors TaTDRL and TaMYB103 Synergistically Activate the Expression of TAA1a in Wheat, Which Positively Regulates the Development of Microspore in Arabidopsis"

_ijms, 2022, doi:10.3390/ijms23147996_

Round 1

Reviewer 1 Report

The manuscript contains the results of the studying the interacting partners involving in transcriptional regulation of TAA1a wheat gene which is important for pollen development. The authors have isolated two transcriptional factor encoding genes, TaTDRL and TaMYB103, and their target gene. In a series of excellent experiments with the use of transgenic Arabidopsis as a model, the synergistic activation of the activity of proTAA1a by TaTDRL and TaMYB103 was demonstrated. The research is relevant for understanding mechanisms of pollen fertility in relation with importance of exploiting the heterosis in wheat crop. The abstract corresponds to the content of the manuscript. The text is illustrated with eight informative figures and added with The Supplementary materials containing both figures and tables.

Comments.

1.       In my opinion, the information on the wheat genotypes used in the study should be extended. It is not clear utilization of the term “near-isogenic lines CMS wheat—MS(KOTS)-90-110 (MS line) and 365 MR(KOTS)-90-110 (MF line)”. Usually, MS and MF lines used in hybrid breeding are designated as “sterile” and “maintainer” lines. Therefore, it is necessary to specify that the near-isogenic lines used in the study differ in the cytoplasm type and have identical nuclear genotypes, including recessive alleles of nuclear fertility restoration genes (Rf). I suggest to include the information in the Introduction section.

2.       A few corrections should be also made in the text:

-           Line 109: expression “microspore fertility” is erroneous. Please correct.

-          Line 124: please replace Figure 2G by Figure 1G.

-          Please decipher the designations in Figure S4 and Figure S5.

-          Line 215: Probably should be (B, G, L) instead of (B, G, I)

-          Line 230: the expression “most was active” as pollen characteristics is unclear.

-          Line 231: Is the reference (Figure 5H, J) right? From the figure content the Figure 5H, M) should be correct.

-          Line 269: The expression “high recoverability” is not clear.

Author Response

Dear Reviewers:

Thanks very much for taking your time to review our manuscript ( ID: ijms-1803715 ). We sincerely thank you for your recognition of our work.Your comments have given us confidence and courage to continue our research along this direction. According to your comments and suggestions, we have carefully revised the manuscript. We have made corresponding modifications to each of your suggestions. Please see the attachment.

Once again, thank you very much for your comments.

Reviewer 2 Report

I would like to note that the manuscript of the authors deserves a very high rating. The experiment is very carefully planned and the conclusions of the authors are fully confirmed by the results of the experiments. Separately, I would like to thank the authors for the clear and concise style of presentation, the article is very easy to read and there is no confusion with the results obtained and their correlation with the theory. My only remark concerns Figure 6, it is not possible to disassemble the Bar and its dimension for fragments A-L.

Author Response

Dear Reviewers:

Thanks very much for taking your time to review our manuscript ( ID: ijms-1803715 ). We sincerely thank you for your recognition of our work. Your comments have given us confidence and courage to continue our research along this direction. We have made modifications to your comments and suggestions on Figure 6 in the manuscript. Please see the attachment for the modification report. Thank you!

Once again, thank you very much for your comments.

Reviewer 3 Report

Nice work, results fully supported by experiments and also by statistics.

 The work is could be accepted, because results bring new inside to microspore development on molecular level. However, actually the role and expression of TAA1 gene is intensively studied, manuscript clearly decribe the role of two transcriptional factors. This could be helpfull not only in plant developmental studies, but also in in vitro studies concerning androgenesis, where TAA1 plays significant role.

I have no requests touching the methodology, tables, figures. The references are appropriate.

Author Response

尊敬的审稿人:

非常感谢您花时间审阅我们的手稿(ID:ijms-1803715)。我们衷心感谢您对我们工作的认可。您的评论给了我们信心和勇气,让我们沿着这个方向继续研究。谢谢!

再次,非常感谢您的意见。